# TRAVERSING CHEMICAL SPACE WITH LATENT POTENTIAL FLOWS

## ABSTRACT

We consider the latent traversal problem in studying and exploring the chemical space with the learned latent space of a generative model. We propose a new framework, ***ChemFlow***, that unified previous molecule manipulation and optimization method with a dynamical system perspective. Specifically, we formulate the problem as learning a vector field that transports the mass of the molecular distribution to the region with desired molecular properties or structure diversity. We also propose several alternative dynamics which exhibit various advantages over previous methods. We validate the efficacy of our proposed methods on both supervised and unsupervised molecule manipulation and optimization scenarios.

## 1 INTRODUCTION

Unveiling the structure of the chemical space and designing efficient search algorithms to explore it is a long-standing challenge in chemistry (Bohacek et al., 1996; Lipinski et al., 2012). Recently, with the promising results by deep generative models to generate valid molecules, more attention has been attracted to study the structure of the learned latent space of those models and optimization methods over the latent space to find better molecule candidates (Du et al., 2022a).

Initially, researchers explicitly introduces additional constraints to encourage a disentangled latent space such that any dimension learns a meaningful factor that may correspond to a molecular property (Du et al., 2022b;c). However, it turns out the assumption is too strong thus cannot learn a meaningful latent space. Inspired by the study of interpretability of generative models on images (Shen et al., 2020), Du et al. (2023) leverage the observed structure similarity and the "linear separability" assumption (Gómez-Bombarelli et al., 2018) to leverage a linear model on the pre-trained model to find meaningful paths in the latent space corresponding to molecular properties. Another branch of work explicitly focus on leveraging the smooth and low-dimensional latent space for molecule optimization with a commonly used approach is gradient ascent over the latent vectors (Liu et al., 2018; Griffiths & Hernández-Lobato, 2020).

In this paper, we propose a new framework, ***ChemFlow***, based on potential flows to efficiently explore the latent structure of molecule generative models. Specifically, we unify previous approaches (gradient-based optimization, linear latent traversal, and disentangled traversal) under the realm of flow that transforms data density along time via a vector field. In contrast to previous linear models, our framework is flexible to learn nonlinear transformations inspired by popular partial differential equations (PDEs) such as heat and wave equations. We also analyze the special properties of each distinct dynamics. For example, under the mild assumption, Langevin dynamics by Fokker Planck equation exhibit convergence to the global minimum. Our framework can also generalized to both supervised and unsupervised settings. Particularly in the unsupervised setting, we introduce a structure diversity potential to find directions that maximize the structure change of the molecules. We conduct extensive experiments with physicochemical properties and drug-related properties on both molecule manipulation and optimization experiments. The experiment results demonstrate the proposed alternative methods using different dynamical priors achieve better or comparable results with existing approaches.

## 2 BACKGROUND: TRAVERSING LATENT SPACE OF MOLECULES

The latent space $\mathcal{Z}$ of molecule generative models is often learned through an encoder function $f_\theta(\cdot)$ and a decoder function $g_\psi(\cdot)$ such that the encoder maps the input molecular structures $\boldsymbol{x} \in \mathcal{X}$ into

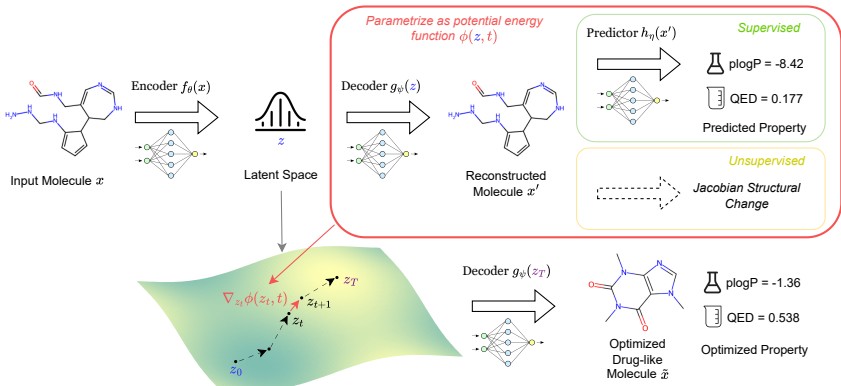

Figure 1: **ChemFlow** framework: (1) a pre-trained encoder $f_\theta(\cdot)$ and decoder $g_\psi(\cdot)$ that maps between molecules $x$ and latent vectors $z$, (2) we use a property predictor $h_\eta(\cdot)$ (green box) or a "Jacobian control" (yellow box) as the guidance to learn a vector field $\nabla\phi(z_t, t)$ that maximizes the change in certain molecular properties (e.g. plogP, QED) or molecular structures, (3) during the training process, we add additional dynamical regularization on the flow. The learned flows move the latent samples to change the structures and properties of the molecules smoothly. (Better seen in color). The flow chart illustrates a case where a molecule is manipulated into a drug-like caffeine.

an (often) low-dimensional and continuous space (i.e. latent space) while the decoder maps the latent vectors $z \in \mathcal{Z}$ back to molecular structures $x'$. Note that this encoder-decoder architecture is general and can be realized by popular generative models such as VAEs, flow-based models, GANs, and diffusion models (Jin et al., 2018; Madhawa et al., 2019; Cao & Kipf, 2018; Vignac et al., 2023). For simplicity, we focus on VAE-based methods in this paper. To traverse the learned latent space of molecule generative models, two approaches have been proposed: gradient-based optimization and latent traversal.

The gradient-based optimization methods first learn a proxy function $h(\cdot)$ parameterized by a neural network that provides the direction to traverse (Zang & Wang, 2020). This can be formulated as a gradient flow following the direction of steepest descent of the potential energy function $h(\cdot)$ and discretized, as follows:

$$
\begin{aligned}
\mathrm{d}z_t &= -\nabla_z h(z_t)\mathrm{d}t \\
z_t &= z_{t-1} - \nabla_z h(z_{t-1})\mathrm{d}t
\end{aligned}
\tag{1}
$$

The latent traversal approaches leverage the observation of linear separability in the learned latent space of molecule generative models (Gómez-Bombarelli et al., 2018). Since the direction is assumed to be linear, it can be found easily. ChemSpace (Du et al., 2023) learns a linear classifier that defines the separation boundary of the molecular properties. Then the normal direction of the boundary provides a linear direction $n \in \mathcal{Z}$ for traversing the latent space:

$$
z_t = z_0 + nt
\tag{2}
$$

We notice that the above gradient flow and linear traversal can be analyzed and designed in a dynamical system perspective, e.g. linear traversal can be considered as a special case of wave functions. This connection inspires us to consider designing more dynamical traversal approches.

We leave a more substantial background for Wasserstein gradient flow and related work in Appendix Sec. A and Sec. E, respectively.

## 3 METHODOLOGY

We present **ChemFlow** as a unified framework for latent traversals in chemical latent space as potential flows (detailed in Appendix A). Motivated by the optimal transport theory of Wasserstein gradient flows, we parameterize a set of scalar potential energies $\phi^k = \mathtt{MLP}_{\theta^k}(z, t) \in \mathbb{R}$ using Physics-informed Neural Networks (PINNs) and take potential flow $\nabla_z \phi$ to traverse the latent sam-

ples:

$$\boldsymbol{z}_t = \boldsymbol{z}_{t-1} + \nabla_{\boldsymbol{z}}\phi^k(\boldsymbol{z}_{t-1}, t-1) \tag{3}$$

**Learning latent potential flows.** Given a pre-trained molecule generative model $g_\psi : \mathcal{Z} \to \mathcal{X}$ with prior distribution $p(\boldsymbol{z})$, we would like to model $K$ different latent trajectories that correspond to different molecule properties. The optimal transport is given by Hamilton-Jacobi Equation (HJE):

$$\frac{\partial}{\partial t}\phi(\boldsymbol{z}, t) + \frac{1}{2}||\nabla_{\boldsymbol{z}}\phi(\boldsymbol{z}, t)||^2 = 0 \tag{4}$$

where the velocity field is defined as the potential flow $\nabla\phi$. The HJE is usually interpreted as fluid-dynamic optimal transport, *i.e.*, under the velocity field $\nabla\phi$, the fluid will evolve to the target distribution with the optimal transport cost. Despite the optimality in moving distributions, HJE may not create smooth trajectories due to the advection field $\nabla\phi$, which may not lead to desired smooth property variations. Alternatively, we can sacrifice the optimal transport property and restrict the advection term to enforce other types of dynamics for smooth spatiotemporal dynamics. For example, we can specify the flow to follow the wave-like dynamics with the coefficient $c$:

$$r^k(\boldsymbol{z}_t, t) = \frac{\partial^2}{\partial t^2}\phi^k(\boldsymbol{z}_t, t) - c^2\nabla_{\boldsymbol{z}}^2\phi^k(\boldsymbol{z}_t, t) \tag{5}$$

The above constraint empirically produces highly diverse and realistic trajectories. We use a PINN (Raissi et al., 2019) to enforce the PDE constraint. Compared with traditional PDE solvers, PINNs can be orders of magnitude faster. Our PINN objective is to minimize:

$$\mathcal{L}_r = \frac{1}{T}\sum_{t=0}^{T-1}||r^k(\boldsymbol{z}_t, t)||_2^2, \ \mathcal{L}_\phi = ||\nabla_{\boldsymbol{z}}\phi^k(\boldsymbol{z}_0, 0)||_2^2 \tag{6}$$

where $T$ represents the total number of traversal steps, $\mathcal{L}_r$ restricts the energy to obey our physical constraints, and $\mathcal{L}_\phi$ restricts $\phi(\boldsymbol{z}_t, t)$ to match the initial condition.

**Supervised semantic potential guidance.** When an explicit semantic potential or labeled data for the semantic of interest is available, we can use the provided semantic potential to guide the learning of the flow. Firstly, we train a surrogate model $h_\eta : \mathcal{X} \to \mathbb{R}$ (parameterized by a deep neural network) to predict the corresponding molecular property. Then we use the trained surrogate model as guidance to learn flows that drive the increase of the property:

$$d = \langle -\nabla_{\boldsymbol{z}}h_\eta(g_\psi(\boldsymbol{z}_t)), \nabla_{\boldsymbol{z}}\phi^k(\boldsymbol{z}_t, t)\rangle, \ \mathcal{L}_\mathcal{P} = -\operatorname{sign}(d)||d||_2^2 \tag{7}$$

The intuition behind this objective is to learn the vector field $\boldsymbol{z}_t$ such that it aligns with the direction of the steepest descent (negative gradient) of the objective function. Note that the sign of the dot product matters as it determines minimizing or maximizing the property.

**Unsupervised structure diversity guidance.** When no explicit potential function is provided to learn the flow, we need to define a potential that captures the change of the molecular properties. As molecular properties are determined by the structures, we devise a potential energy that maximizes the continuous structure change of the generated molecules. Inspired by Song et al. (2023b), we couple the traversal direction with the Jacobian of the generator to maximize the traversal variations in the molecular space. We therefore introduce the guidance $\mathcal{L}_\mathcal{J}$ as follows:

$$g(\boldsymbol{z}_t + \epsilon\nabla_{\boldsymbol{z}}\phi^k(\boldsymbol{z}_t, t)) \approx g(\boldsymbol{z}_t) + \epsilon\underline{\frac{\partial g(\boldsymbol{z}_t)}{\partial \boldsymbol{z}_t}\nabla_{\boldsymbol{z}}\phi^k(\boldsymbol{z}_t, t)}, \mathcal{L}_\mathcal{J} = -\left\|\frac{\partial g(\boldsymbol{z}_t)}{\partial \boldsymbol{z}_t}\nabla_{\boldsymbol{z}}\phi^k(\boldsymbol{z}_t, t)\right\|_2^2 \tag{8}$$

where we perform the first-order Taylor approximation on the left equation. In the unsupervised setting, for sufficiently small $\epsilon$, if the Jacobian-vector product (the underlined term in Eq. (8)) can cause large variations in the generated sample, the direction is likely to correspond to certain properties of molecules. Compared to the supervised setting which maximizes the change of the molecular properties, it aims to find the direction that causes the maximal change of the structures. This can in turn effectively push the initial data distribution to the target one concentrated on the maximum property value. The Jacobian guidance will compete with the dynamical regularization (e.g. wave-like form) on the flow to yield smooth and meaningful traversal paths. In high-dimensional space, a trivial solution exists such that all the flows learn the exact same direction. To encourage the model to learn more disentangled paths, we design a disentanglement regularization detailed in Appendix B.

In Appendix C, we show one particular property of a specific flow governed by Fokker Planck equation such that it is equivalent to simulating Langevin dynamics in the latent space which has weak convergence to the global minimizer of the molecular property.

## 4 EXPERIMENTS

We conduct experiments by pre-training a VAE model on a commonly used molecular dataset (*ZINC250k* (Irwin & Shoichet, 2005)). We compare our proposed methods against two baseline methods with linear traversal (Du et al., 2023) and gradient flow-based traversal (Eckmann et al., 2022). For all other details, we defer to Appendix F.

Due to the space limit, we defer all experiments related to molecule manipulation in Appendix F.5. In the main paper, we only report results for molecule optimization in the latent space.

| $\delta$ | Random | Random-1D | ChemSpace | Gradient Flow | Wave (SPV) | Wave (UNSUP) | HJ (SPV) | HJ (UNSUP) | LD |
|---|---|---|---|---|---|---|---|---|---|
| 0 | 45.5 ± 13.3 (99.5) | 13.4 ± 12.3 (19.5) | 49.9 ± 12.2 (99.5) | 30.6 ± 17.9 (87.8) | 13.8 ± 12.9 (30.9) | 41.7 ± 18.8 (97.8) | 48.7 ± 12.3 (99.6) | 40.8 ± 19.3 (96.6) | **57.1 ± 11.3 (100.0)** |
| 0.2 | 20.7 ± 14.4 (81.8) | 11.9 ± 11.1 (18.0) | 26.3 ± 16.8 (86.9) | 23.2 ± 16.2 (77.8) | 12.3 ± 10.8 (28.7) | 18.0 ± 14.7 (68.6) | 23.2 ± 16.1 (82.2) | 16.2 ± 13.4 (70.1) | **32.0 ± 16.3 (96.4)** |
| 0.4 | 12.8 ± 10.7 (57.0) | 9.9 ± 10.1 (14.9) | 15.4 ± 13.9 (61.8) | 13.9 ± 11.8 (57.6) | 9.6 ± 9.2 (22.9) | 9.4 ± 9.2 (41.9) | 13.2 ± 12.0 (56.2) | 9.0 ± 9.1 (46.5) | **16.3 ± 12.4 (77.4)** |
| 0.6 | 8.0 ± 8.3 (29.5) | 6.4 ± 8.0 (7.9) | 9.5 ± 10.0 (30.0) | 9.2 ± 8.8 (31.8) | 6.9 ± 7.1 (13.9) | 6.5 ± 7.0 (24.1) | 8.3 ± 8.1 (28.7) | 6.3 ± 7.4 (25.8) | **9.8 ± 9.3 (47.0)** |

Table 1: **Similarity-constrained QED maximization.** The value of QED is scaled by 100 for better presentation.

**Constrained molecule optimization.** Molecule optimization is one of the fundamental problems in drug and materials discovery where we aim to find molecules with better properties (Brown et al., 2019). Following the procedures described in JT-VAE (Jin et al., 2018) and LIMO (Eckmann et al., 2022), we select the 800 molecules with the lowest QED scores in the ZINC250k dataset and perform 1,000 steps of optimization until all methods are converged. Table 1 reports the statistics of QED improvements under different similarity constraints. Among all the approaches, Langevin dynamics achieves the best overall performance. As we analyze above, it is more suitable for the optimization task than other dynamical priors. In addition, we further show it enjoys a faster convergence rate empirically than other dynamical priors as well in Figure 5. Surprisingly, we observe that the random direction performs also quite well on molecule optimization tasks. This observation motivates us to study the structure of the latent space. We show that the molecular structure distribution on the latent space follows a high-dimensional Gaussian distribution and the random direction increases the norm of the latent vectors that have strong correlations with molecular properties. We analyze this systematically in Appendix G. It is also notable that although random directions could be effective in optimizing molecules, the distribution of the entire molecule sets being optimized does not change accordingly as shown in Figure 2.

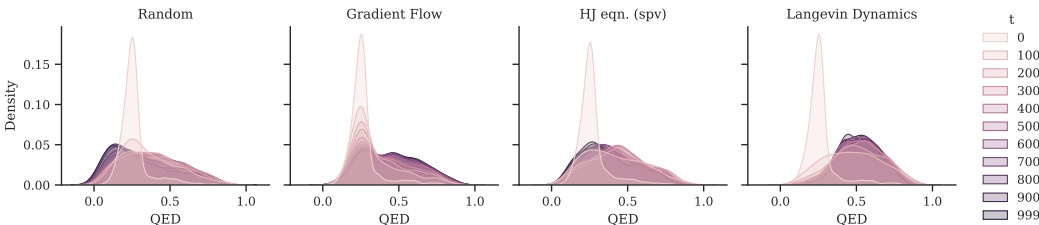

Figure 2: **Molecular property distribution shifts following the latent traversal path.**

We also report additional experiment on unconstrained and multi-objective molecule optimization results in Appendix F.7 and F.8, respectively.

## 5 CONCLUSION, LIMITATION AND FUTURE WORK

In this paper, we propose a unifying framework that learns a flow transformation through a vector field for traversing the latent space of molecule generative models. Under this framework, we propose a variety of dynamical regularizations which exhibit different properties. We hope this unifying framework can open up a new research avenue to study the structure and dynamics of the latent space of molecule generative models.

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

# Appendix for ChemFlow

## A WASSERSTEIN GRADIENT FLOW

Gradient flows define the curve $\boldsymbol{x}(t) \in \mathbb{R}^n$ that evolves in the direction of steepest descent of a function $\mathcal{F} : \mathbb{R}^n \to \mathbb{R}$. The time evolution of the gradient flow is given by the ODE $\boldsymbol{x}'(t) = -\nabla \mathcal{F}(\boldsymbol{x}(t))$. The choice of the functional $\mathcal{F}$ determines the metric space and the associated geodesics. Wasserstein gradient flows describe a special type of gradient flow where $\mathcal{F}$ is set to be the Wasserstein distance. For example, as introduced in Benamou & Brenier (2000), the commonly used $L_2$ Wasserstein distance has the following dynamic formulation:

$$W_2(\rho_0, \rho_1)^2 = \min_{\rho, v} \left\{ \int \int \frac{1}{2} \rho(\boldsymbol{x}, t) |v(\boldsymbol{x}, t)|^2 \, dx \, dt : \partial_t \rho(\boldsymbol{x}, t) = -\nabla \cdot (v(\boldsymbol{x}, t) \rho(\boldsymbol{x}, t)) \right\} \quad (9)$$

where $\rho_0$ and $\rho_1$ are two probability measures at the source and target distributions, respectively. Interestingly, if we take the gradient of a potential energy $\nabla \phi$ as the velocity field applied to a distribution, the time evolution of $\nabla \phi$ can be seen to minimize the Wasserstein distance and thus follow optimal transport.

In Appendix A, we give detailed derivations of how the vector fields minimize the $L_2$ Wasserstein distance and discuss some PDEs of the density evolution that can be interpreted under different learned potentials.

As shown in the main paper, based on the dynamic formulation of optimal transport (Benamou & Brenier, 2000), the $L_2$ Wasserstein distance can be re-written as:

$$W_2(\rho_0, \rho_1) = \min_{\rho, v} \sqrt{\int \int \rho_t(\boldsymbol{z}) |v_t(\boldsymbol{z})|^2 \, d\boldsymbol{z} dt} \tag{10}$$

where $v_t(\boldsymbol{z})$ is the velocity of the particle at position $\boldsymbol{z}$ and time $t$, and $\rho_t(\boldsymbol{z})$ is the density $d\mu(\boldsymbol{z}) = \rho_t(\boldsymbol{z})d\boldsymbol{z}$. The distance can be optimized by the gradient flow of a certain function on space and time. Consider the functional $\mathcal{F} : \mathbb{R}^n \to \mathbb{R}$ that takes the following form:

$$\mathcal{F}(\mu) = \int U(\rho_t(\boldsymbol{z})) \, d\boldsymbol{z} \tag{11}$$

The curve is considered as a gradient flow if it satisfies $\nabla \mathcal{F} = -\frac{d}{dt}\rho_t(\boldsymbol{z})$ (Ambrosio et al., 2005). Moving the particles leads to:

$$\frac{d}{dt}\mathcal{F}(\mu) = \int U'(\boldsymbol{z})\frac{d\,\rho_t(\boldsymbol{z})}{dt} \, d\boldsymbol{z} \tag{12}$$

The velocity vector satisfies the continuity equation:

$$\frac{d\,\rho_t(\boldsymbol{z})}{dt} = -\nabla \cdot \left( v_t(\boldsymbol{z})\rho_t(\boldsymbol{z}) \right) \tag{13}$$

where $-\nabla \cdot \left( v_t(\boldsymbol{z})\rho_t(\boldsymbol{z}) \right)$ is the tangent vector at point $\rho_t(\boldsymbol{z})$. Eq. (12) can be simplified to:

$$\begin{aligned}
\frac{d}{dt}\mathcal{F}(\mu) &= \int -U'(\rho_t(\boldsymbol{z}))\nabla \cdot \left( v_t(\boldsymbol{z})\rho_t(\boldsymbol{z}) \right) d\boldsymbol{z} \\
&= \int \nabla\left( U'(\rho_t(\boldsymbol{z})) \right) v_t(\boldsymbol{z})\rho_t(\boldsymbol{z}) \, d\boldsymbol{z}
\end{aligned} \tag{14}$$

On the other hand, the calculus of differential geometry gives

$$\frac{d}{dt}\mathcal{F}(\mu) = \mathrm{Diff}\mathcal{F}|_{\rho_t}\left(-\nabla \cdot \left( v_t(\boldsymbol{z})\rho_t(\boldsymbol{z}) \right)\right) = \langle \nabla \mathcal{F}, -\nabla \cdot \left( v_t(\boldsymbol{z})\rho_t(\boldsymbol{z}) \right) \rangle_f \tag{15}$$

where $\langle, \rangle_f$ is a Riemannian distance function which is defined as:

$$\langle -\nabla \cdot \left( w_1(\boldsymbol{z})\rho_t(\boldsymbol{z}) \right), -\nabla \cdot \left( w_2(\boldsymbol{z})\rho_t(\boldsymbol{z}) \right) \rangle_f = \int w_1(\boldsymbol{z})w_2(\boldsymbol{z})f(\boldsymbol{z}) \, d\boldsymbol{z} \tag{16}$$

This scalar product coincides with the $W_2$ distance according to Benamou & Brenier (2000). Then Eq. (14) can be similarly re-written as:

$$\frac{d}{dt}\mathcal{F}(\mu) = \langle -\nabla \cdot \left( \nabla U'(\rho_t(\boldsymbol{z}))\rho_t(\boldsymbol{z}) \right), -\nabla \cdot \left( v_t(\boldsymbol{z})\rho_t(\boldsymbol{z}) \right) \rangle_f \tag{17}$$

So the relation arises as:

$$\nabla \mathcal{F} = -\nabla \cdot \left( \nabla U'(\rho_t(\boldsymbol{z}))\rho_t(\boldsymbol{z}) \right) \tag{18}$$

Since we have $\nabla \mathcal{F} = -\frac{d}{dt}\rho_t(\boldsymbol{z})$, the above equation can be re-written as

$$\frac{d}{dt}\rho_t(\boldsymbol{z}) = \nabla \cdot \left( \nabla U'(\rho_t(\boldsymbol{z}))\rho_t(\boldsymbol{z}) \right) \tag{19}$$

The above derivations can be alternatively made by JKO schemes (Jordan et al., 1996). This explicitly defines the relation between evolution PDEs of $\rho_t(\boldsymbol{z})$ and the internal energy $U$. For our method, we use the gradient of our scalar energy field $\nabla u(\boldsymbol{z}, t)$ to learn the velocity field which is given by $U'(\rho_t(\boldsymbol{z}))$. Interestingly, driven by certain specific velocity fields $\nabla u(\boldsymbol{z}, t)$, the evolution of $\rho(\boldsymbol{z}, t)$ would become some special PDEs. Here we discuss some possibilities:

**Heat Equations.** If we consider the energy function $U$ as the weighted entropy:

$$U(\rho_t(\boldsymbol{z})) = \rho_t(\boldsymbol{z})\log(\rho_t(\boldsymbol{z})) \tag{20}$$

We would have exactly the heat equation:

$$\frac{d}{dt}\rho_t(\boldsymbol{z}) - \frac{d}{d\boldsymbol{z}^2}\rho_t(\boldsymbol{z}) = 0 \tag{21}$$

Injecting the above equation back into the continuity equation leads to the velocity field $v_t(\boldsymbol{z})$ as

$$\frac{d\,\rho_t(\boldsymbol{z})}{dt} = -\nabla \cdot \Big(v_t(\boldsymbol{z})\rho_t(\boldsymbol{z})\Big) = \frac{d}{d\boldsymbol{z}^2}\rho_t(\boldsymbol{z})$$
$$v_t(\boldsymbol{z}) = -\frac{\nabla\rho_t(\boldsymbol{z})}{\rho_t(\boldsymbol{z})} = -\nabla\log(\rho_t(\boldsymbol{z})) \tag{22}$$

When our $\nabla u(\boldsymbol{z},t)$ learns the velocity field $-\nabla\log(\rho_t(\boldsymbol{z}))$, the evolution of $\rho(\boldsymbol{z},t)$ would become heat equations.

**Fokker Planck Equations.** For the energy function defined as:

$$U(\rho_t(\boldsymbol{z})) = -A \cdot \rho_t(\boldsymbol{z}) + \rho_t(\boldsymbol{z})\log(\rho_t(\boldsymbol{z})) \tag{23}$$

we would have the Fokker-Planck equation as

$$\frac{d}{dt}\rho_t(\boldsymbol{z}) + \frac{d}{d\boldsymbol{z}}[\nabla A\rho_t(\boldsymbol{z})] - \frac{d}{d\boldsymbol{z}^2}[\rho_t(\boldsymbol{z})] = 0, \tag{24}$$

The velocity field can be similarly derived as

$$v_t(\boldsymbol{z}) = \nabla A - \nabla\log(\rho_t(\boldsymbol{z})) \tag{25}$$

For the velocity field $\nabla A - \nabla\log(\rho_t(\boldsymbol{z}))$, the movement of $\rho(\boldsymbol{z},t)$ is the Fokker Planck equation.

**Porous Medium Equations.** If we define the energy function as

$$U(\rho_t(\boldsymbol{z})) = \frac{1}{m-1}\rho_t^m(\boldsymbol{z}) \tag{26}$$

Then we would have the porous medium equation where $m > 1$ and the velocity field:

$$\frac{d}{dt}\rho_t(\boldsymbol{z}) - \frac{d}{d\boldsymbol{z}^2}\rho_t^m(\boldsymbol{z}) = 0, \ v_t(\boldsymbol{z}) = -m\rho^{m-2}\nabla\rho \tag{27}$$

When the $\nabla u(\boldsymbol{z},t)$ learns the velocity $-m\rho^{m-2}\nabla\rho$, the trajectory of $\rho(\boldsymbol{z},t)$ becomes the porous medium equations.

## B  ADDITIONAL METHOD DETAILS

**Disentanglement Regularization.** While the above formulation can encourage smooth dynamics and meaningful output variations, the potential flows are likely to mine identical directions which all correspond to the maximum Jacobian change. To avoid such a trivial solution, we adopt an auxiliary classifier $l_\gamma$ to predict the potential index and use the cross-entropy loss to optimize it:

$$\hat{k}=l_\gamma(\boldsymbol{z}_t; \boldsymbol{z}_{t+1}), \ \mathcal{L}_k = \mathcal{L}_{CE}(\hat{k}, k) \tag{28}$$

Where $\boldsymbol{x}_t = g(\boldsymbol{z}_t)$ is the generated sample from timestep $t$. We see the extra classifier guidance would encourage each potential flow to be independent and find distinct properties.

## C  CONNECTION WITH LANGEVIN DYNAMICS FOR GLOBAL OPTIMIZATION

In scenarios where our flow adheres to the dynamics of the Fokker-Planck equation, our approach may also be interpreted as employing a learned potential energy function to simulate Langevin Dynamics for global optimization (Gardiner et al., 1985). Notably, the convergence of Langevin dynamics, particularly at low temperatures, tends to occur around the global minimum of the potential energy function (Chiang et al., 1987). The continuous and discretized Langevin dynamics are as follows:

$$\mathrm{d}\boldsymbol{z}_t = -\nabla_z h_\eta(\boldsymbol{z}_t)\mathrm{d}t + \sqrt{2}\mathrm{d}\mathbf{w}_t$$
$$\boldsymbol{z}_t = \boldsymbol{z}_{t-1} - \nabla_z h_\eta(\boldsymbol{z}_{t-1})\mathrm{d}t + \sqrt{2\mathrm{d}t}\mathcal{N}(0, I) \tag{29}$$

**Proposition C.1.** *(Global Convergence of Langevin Dynamics, adapted from Gelfand & Mitter (1991)). Given a langevin dynamics in the form of*

$$\boldsymbol{z}_t = \boldsymbol{z}_{t-1} - a_t(\nabla_z h_\eta(\boldsymbol{z}_{t-1}) + \boldsymbol{u}_t) + b_t\mathbf{w}_t \tag{30}$$

*where $\mathbf{w}_t$ is a d-dimensional Brownian motion, $a_t$ and $b_t$ are a set of positive numbers with $a_T, b_T \to 0$, and $\boldsymbol{u}_t$ is a set of random variables in $\mathbb{R}^n$ denoting noisy measurements of the energy function $h_\eta(\cdot)$. Under mild assumptions, $\boldsymbol{z}_t$ converges to the set of global minima of $h_\eta(\cdot)$ in probability.*

Following Proposition C.1, we show that the learned latent potential can be used to search for molecules with optimal properties and it converges to the global minima of the learned latent potential.

## D    ALGORITHMS

Due to space limit, we demonstrate the training and inference procedure for ***ChemFlow*** in Alg. 2 and Alg. 2, respectively.

---
**Algorithm 1** ChemFlow Inference / Traversal
---
**Require:** Pre-trained encoder $f_\theta$, pre-trained potential function $\phi$, (optional) pre-trained proxy function $h$, timestamps $T$, step size $\alpha$, LD strenth $\beta$
1: Sampling: $\boldsymbol{z}_0 = f_\theta(x_0)$
2: **for** $t = 1, \dots, T$ **do**
3:      **if** Langevin Dynamics **then**
4:          $\boldsymbol{z}_t = \boldsymbol{z}_{t-1} - \alpha\nabla_{\boldsymbol{z}}h_\eta(\boldsymbol{z}_{t-1}) + \beta\sqrt{2\alpha}\mathcal{N}(0, I)$
5:      **else if** Gradient Flow **then**
6:          $\boldsymbol{z}_t = \boldsymbol{z}_{t-1} - \alpha\nabla_{\boldsymbol{z}}h_\eta(\boldsymbol{z}_{t-1})$
7:      **else**
8:          $\boldsymbol{z}_t = \boldsymbol{z}_{t-1} + \alpha\nabla_{\boldsymbol{z}}\phi(\boldsymbol{z}_{t-1}, t - 1)$
9:      **end if**
10: **end for**
---

---
**Algorithm 2** ChemFlow Training
---
**Require:** Pre-trained encoder $f_\theta$, decoder $g_\psi$, (optional) classifier $l_\gamma$, timestamps $T$, # of potential functions $K$
1: Initialize $\phi^j(\cdot) \leftarrow$ MLP for $j = 1, \dots, K$
2: **repeat**
3:      Sampling: $\boldsymbol{z}_0 = f_\theta(x_0), t \sim \texttt{Categorical}(T), k \sim \texttt{Categorical}(K)$
4:      **for** $i = 1, \dots, t$ **do**
5:          $\boldsymbol{z}_{i+1} = \boldsymbol{z}_i + \nabla_{\boldsymbol{z}}\phi^k(\boldsymbol{z}_i, i)$
6:      **end for**
7:      Decode: $\boldsymbol{x}_t = g_\psi(\boldsymbol{z}_t), \boldsymbol{x}_{t+1} = g_\psi(\boldsymbol{z}_{t+1})$
8:      **if** unsupervised **then**
9:          Classification: $\hat{k} = l_\gamma(\boldsymbol{x}_t; \boldsymbol{x}_{t+1})$
10:          Loss: $\mathcal{L} = \mathcal{L}_r + \mathcal{L}_\phi + \mathcal{L}_{\mathcal{J}} + \mathcal{L}_k$
11:      **else**
12:          Loss: $\mathcal{L} = \mathcal{L}_r + \mathcal{L}_\phi + \mathcal{L}_{\mathcal{P}}$
13:      **end if**
14:      Back-propagation through the Loss $\mathcal{L}$
15: **until** converged
---

## E    RELATED WORK

### E.1    MACHINE LEARNING FOR MOLECULE GENERATION

Molecules are highly discrete objects and two branches of methods are thus developed to design or search new molecules (Du et al., 2022a). One idea is to leverage the advancement of deep generative models which approximate the data distribution from a provided dataset of molecules and then sample new molecules from the learned density. This idea inspires a line of work developing deep generative models from variational auto-encoders (VAE) (Gómez-Bombarelli et al., 2018; Jin et al., 2018), generative adversarial networks (GAN) (Guimaraes et al., 2017; Cao & Kipf, 2018), normalizing flows (NF) (Madhawa et al., 2019; Zang & Wang, 2020) and more recently diffusion models (Hoogeboom et al., 2022; Vignac et al., 2023; Jo et al., 2022). However, to respect the combinatorial nature of molecules, another line of work leverage combinatorial optimization to search new

molecules including genetic algorithm (GA) (Jensen, 2019), Monte Carlo tree search (MCTS) (Yang et al., 2017), reinforcement learning (RL) (You et al., 2018), but often with sophisticated optimization objectives beyond simple valid molecules.

### E.2 GOAL-ORIENTED MOLECULE GENERATION

In addition to simply generating valid molecules, a more realistic application is to generate molecules with desired properties (Du et al., 2022a). For deep generative model-based methods, it is naturally combined with on-the-fly optimization methods such as gradient-based or Bayesian optimization (in low data regime) as it often maps data to a low-dimensional and smooth latent space thus more friendly for these optimization methods (Griffiths & Hernández-Lobato, 2020). For methods that do not explicitly reduce the dimensionality of data such as diffusion models, Schneuing et al. (2022) propose an evolutionary process to iteratively optimize the generated molecules. As it is observed that the learned latent space exhibits explicit structure (Gómez-Bombarelli et al., 2018), Du et al. (2023) leverage such property to learn a linear classifier to find the latent direction to optimize the property of given molecules. In opposition to deep generative models, combinatorial optimization methods are often inherently associated with optimization, e.g. reward function in RL, selection criteria in GA, etc (Fu et al., 2022; Loeffler et al., 2023).

### E.3 LATENT TRAVERSAL FOR SEMANTIC IMAGE EDITING

Beyond molecule generation, there is a vast literature on the study of the latent space of generative models on images for image editing and manipulation (Goetschalckx et al., 2019; Jahanian et al., 2020; Voynov & Babenko, 2020; Härkönen et al., 2020; Zhu et al., 2020; Peebles et al., 2020; Shen & Zhou, 2021; Song et al., 2022; 2023b;a;c). Here we highlight some representative supervised and unsupervised approaches. Supervised methods usually require pixel-wise annotations. Interface-GAN (Shen et al., 2020) leverages face image pairs of different attributes to interpret disentangled latent representations of GANs. Jahanian et al. (2020) explores linear and non-linear walks in the latent space under the guidance of user-specified transformation. Compared to supervised methods, unsupervised ones mainly focus on discovering meaningful interpretable directions in the latent space through extra regularization. Voynov & Babenko (2020) proposes to jointly learn a set of orthogonal directions and a classifier to learn the distinct interpretable directions. SeFa (Shen & Zhou, 2021) and HouseholderGAN (Song et al., 2023c) propose to use the eigenvectors of the (orthogonal) projection matrices as interpretable directions to traverse the latent space. More relevantly, Song et al. (2023b) proposes to use wave-like potential flows to model the spatiotemporal dynamics in the latent spaces of different generative models.

## F EXPERIMENTS DETAILS

### F.1 BASELINES

We compare with the following baselines:

- Random: we take a linear direction that is sampled from Multi-variant Gaussian distribution in the high dimensional latent space and normalized to unit length for all molecules across all time steps.

- Random 1D: we take a unit vector where only 1 randomly selected dimension is either 1 or -1 as the linear direction.

- ChemSpace (Du et al., 2023): a separation boundary of the training dataset in latent space w.r.t. the desired property is classified by an Support vector machine (SVM). Then we take the normal vector corresponding to the positive separation as the manipulation direction of control.

- LIMO (Eckmann et al., 2022) / Gradient Flow: a VAE-based generative model that encodes the input molecules into SELFIES (Krenn et al., 2019) and auto-regressive on the tokenized molecule. LIMO uses Adam optimizer to reverse optimize on the input latent vector $z$ whereas Gradient Flow is equivalent to using an SGD optimizer for the same purpose.

### F.2 MOLECULE PROPERTIES

We report the following metrics for our experiments:

- **Penalized logP**: Estimated octanol-water partition coefficient penalized by synthetic accessibility (SA) score and the number of atoms in the longest ring.

#### F.2.1 MISALIGNMENT OF NORMALIZATION SCHEMES FOR PENALIZED LOGP

We noticed that plogP is a commonly reported metric in recent molecule discovery literature but does not share the same normalization scheme. Following Gómez-Bombarelli et al. (2018, Eq. 1), the SA scores and a ring penalty term were introduced into the calculation of penalized logP as the following

$$J^{\mathrm{logP}}(m) = \mathrm{logP}(m) - \mathrm{SA}(m) - \mathrm{ring\text{-}penalty}(m)$$

Each term of $\mathrm{logP}(m)$, $\mathrm{SA}(m)$, and $\mathrm{ring\text{-}penalty}(m)$ are normalized to have zero mean and unit standard derivation across the training data. However, no sufficient details were included in their paper or their released source code on how the $\mathrm{ring\text{-}penalty}(m)$ is computed. Specifically, 3 implementations are widely used in various works.

**Penalized by the length of the maximum cycle without normalization** where $\mathrm{ring\text{-}penalty}(m)$ is computed as the number of atoms on the longest ring - 6 in their implementation. Neither $\mathrm{logP}(m)$, $\mathrm{SA}(m)$, or $\mathrm{ring\text{-}penalty}(m)$ is normalized.

**Penalized by the length of the maximum cycle with normalization** where $\mathrm{ring\text{-}penalty}(m)$ is computed same as without normalization. We report plogP using this metric.

**Penalized by number of cycles** As described by Jin et al. (2018), $\mathrm{ring\text{-}penalty}(m)$ is computed as the number of rings in the molecule that has more than 6 atoms. LIMO reports plogP using this metric.

### F.3 EXPERIMENTS SETUP

**Datasets.** We consider a commonly used molecular dataset to study drug-related properties, *ZINC250k* which is a subset of the ZINC database (Irwin & Shoichet, 2005) containing $\sim$250,000 commercially available compounds for virtual screening. The molecules in the dataset is firstly encoded as SELFIES strings for molecule representations and then tokenized into input IDs following the language modeling commons with `[PAD]` and other special tokens. All input IDs are padded to the maximum length in the overall dataset.

**Implementations.** We establish our framework by pre-training a VAE model that learns a latent space of molecules and is capable of generating new molecules by decoding latent vectors from the latent space. We adapt the framework in Eckmann et al. (2022) which is a basic VAE architecture with molecular SELFIES string representations plusing an MLP for surrogate property predictor. See Appendix F.3 for all implementation and hyper-parameter details.

**Model variants.** As discussed in Section 3, our proposed framework is general to incorporate different dynamical priors to learn the flow. For the experiments, we consider four types of dynamics including *gradient flow (GF)*, *Wave flow (Wave, Eq. (5))*, *Hamilton Jacobi flow (HJ, Eq. (4))* and *Langevin Dynamics or equivalently Fokker Planck flow (LD, Eq. (29))*.

**Pre-trained VAE** We adjust the VAE architecture from LIMO that consisting a 64-dimension embedding, 1024 latent space size, 3-hidden-layer encoder, and 3-hidden-layer decoder both with 1D batch normalization and ReLU activation functions. The hidden layer sizes are $\{2000, 1000, 1000\}$ for the encoder and reversely for the decoder. We empirically find that replacing the ReLU activation function with its newer variant Mish activation function (Misra, 2020) results in faster convergence and better validation loss. All the experiments reported in this paper use this Mish-activated variant of VAE. The VAE is trained using an AdamW (Loshchilov & Hutter, 2017) optimizer with default settings of PyTorch implementation, 0.001 initial learning rate, and 1,024 training batch. To better

prevent the model from being stacked at a sub-optimal local minimum, a cosine learning rate scheduler with periodic restart is applied. The VAE is trained for 150 epochs with 4 restarts on 90% of the ZINC250k dataset and validated with the rest 10% data. The checkpoint corresponding to the best validation loss epoch is selected.

**Surrogate Predictor**    Similarly, we find that the Mish activation function improves the performance of the original MLP predictor described by LIMO. The predictor consists of 3 linear hidden layers of size 1024 with Mish activation function. Similar to the LIMO setups, we find that the choice of optimizer and training hyperparameters like learning rate or learning rate scheduler is crucial for successful training. Like LIMO, we use PyTorch Lightning to choose the optimal learning rate for the Adam optimization. The predictor is trained for 20 epochs on 100,000 randomly generated samples and validated with 10,000 unseen data. The epoch with the best validation loss is selected.

**PDE PINN**    We use an MLP structure to parameterize the potential energy function. The spatial time input $t$ is embedded with a sinuous positional embedding followed by a linear layer. The spacial input $x$ is encoded with a linear layer and ReLU activation function. We empirically find that the Tanh and GELU activation function of MLP does not help PINN learn the corresponding Jacobian structure. Therefore, a ReLU activation function is used instead. The training of PINN follows uses 90,000 random data and 10,000 unseen data for validation. For unsupervised settings, 10 disentangled potential energy functions are trained for 70 epochs with a batch size 100. The epoch with the best validation loss is selected.

**Reproducibility**    All the experiments including baselines are conducted on one RTX 3090 GPU and one Nvidia A100 GPU. The code implementation will be released upon the acceptance of the paper.

### F.4    MORE EXPERIMENT RESULTS

We conduct more experiments to analyze the performance of the proposed methods systematically. They are referred and discussed in the main paper.

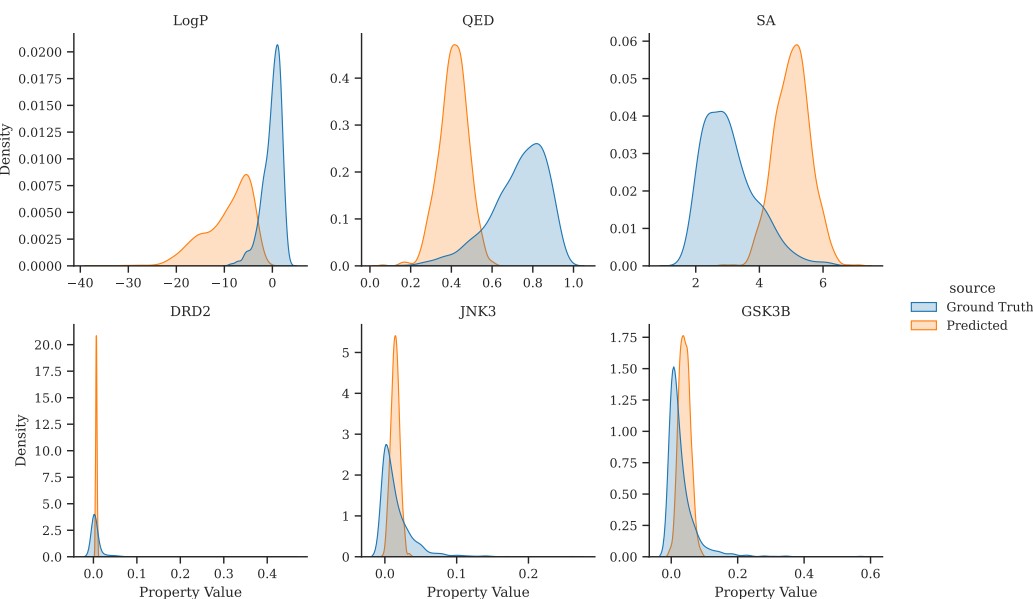

Figure 3: **Distribution for predicted properties and the ground truth.** We hypothesize there is a training and generalization errors in the surrogate model. We have observed the distribution of predicted and ground truth property values are different.

|  | PLOGP | SA | QED | DRD2 | JNK3 | JSK3B |
|---|---|---|---|---|---|---|
| 0 | **0.138** | 0.334 | -0.055 | 0.099 | **0.198** | **0.284** |
| 1 | 0.012 | 0.096 | 0.214 | -0.041 | -0.110 | -0.162 |
| 2 | -0.097 | 0.177 | 0.308 | **0.283** | -0.060 | -0.087 |
| 3 | -0.008 | **0.404** | 0.321 | 0.135 | -0.114 | -0.179 |
| 4 | 0.022 | 0.028 | **0.332** | 0.045 | -0.110 | -0.173 |
| 5 | 0.043 | 0.005 | 0.167 | -0.115 | -0.108 | -0.192 |
| 6 | 0.076 | -0.018 | 0.280 | 0.025 | -0.129 | -0.215 |
| 7 | 0.005 | 0.039 | 0.099 | -0.125 | -0.091 | -0.124 |
| 8 | 0.010 | 0.006 | 0.320 | 0.061 | -0.134 | -0.162 |
| 9 | 0.043 | 0.189 | 0.248 | 0.038 | -0.111 | -0.137 |
| INDEX | 0 | 3 | 4 | 2 | 0 | 0 |

Table 2: **Pearson Correlation.** The average Pearson correlation between the sequence of real properties and sequence of time steps along the manipulation trajectory following a learned potential function $\phi^k(z, t)$ using wave equations.

|  | PLOGP | SA | QED | DRD2 | JNK3 | GSK3B |
|---|---|---|---|---|---|---|
| 0 | 0.062 | 0.176 | 0.105 | -0.107 | -0.066 | -0.069 |
| 1 | 0.086 | -0.027 | 0.291 | -0.017 | -0.125 | -0.160 |
| 2 | 0.009 | 0.131 | -0.018 | -0.008 | 0.007 | 0.010 |
| 3 | -0.008 | -0.028 | 0.123 | -0.065 | -0.038 | -0.052 |
| 4 | 0.052 | 0.032 | 0.254 | -0.027 | -0.095 | -0.197 |
| 5 | **0.199** | **0.416** | -0.310 | -0.233 | **0.150** | **0.228** |
| 6 | 0.036 | 0.002 | 0.292 | **0.019** | -0.103 | -0.134 |
| 7 | -0.055 | 0.392 | 0.035 | 0.005 | 0.009 | 0.023 |
| 8 | 0.046 | -0.04 | 0.296 | -0.037 | -0.127 | -0.158 |
| 9 | 0.051 | -0.007 | **0.320** | -0.075 | -0.115 | -0.155 |
| INDEX | 5 | 5 | 9 | 6 | 5 | 5 |

Table 3: **Pearson Correlation.** The average Pearson correlation between the sequence of real properties and sequence of time steps along the manipulation trajectory following a learned potential function $\phi^k(z, t)$ using Hamilton Jacobi equations.

| $\delta$ | Random | Random-1D | ChemSpace | Gradient Flow | Wave (spv) | Wave (unsup) | HJ (spv) | HJ (unsup) | LD |
|---|---|---|---|---|---|---|---|---|---|
| 0 | 11.28 ± 7.46 (**98.0**) | 5.50 ± 7.26 (23.6) | 11.05 ± 7.66 (96.8) | 10.81 ± 8.12 (94.8) | 10.75 ± 7.99 (96.6) | 9.88 ± 9.89 (92.5) | 10.64 ± 11.13 (86.9) | 9.81 ± 9.48 (89.0) | **11.65 ± 7.96** (97.1) |
| 0.2 | 7.28 ± 6.40 (71.6) | 4.90 ± 6.56 (22.1) | **7.77 ± 6.75** (78.1) | 7.12 ± 6.18 (**72.0**) | 7.03 ± 6.82 (71.5) | 6.15 ± 7.39 (46.9) | 6.41 ± 7.72 (71.1) | 4.60 ± 6.22 (30.9) | 7.24 ± 6.07 (**72.0**) |
| 0.4 | 4.66 ± 5.56 (42.5) | 3.67 ± 5.49 (16.8) | **5.36 ± 5.84** (48.0) | 4.35 ± 4.86 (39.5) | 4.29 ± 5.36 (44.1) | 2.39 ± 3.56 (21.6) | 3.15 ± 5.05 (**49.1**) | 1.53 ± 2.66 (9.2) | 4.44 ± 4.94 (38.8) |
| 0.6 | 2.90 ± 4.67 (18.2) | 1.81 ± 3.71 (9.0) | 3.54 ± 4.46 (20.9) | 2.53 ± 3.52 (11.9) | 1.81 ± 3.26 (18.9) | 1.00 ± 1.16 (10.1) | 1.14 ± 1.83 (24.8) | 0.56 ± 0.36 (3.1) | 2.79 ± 3.80 (12.9) |

Table 4: **Similarity-constrained plogp maximization.** For each method with minimum similarity constraint $\delta$, the results in reported in format mean ± standard derivation (success rate %) of absolute improvement, where the mean and standard derivation are calculated among molecules that satisfy the similarity constraint. The value of QED is scaled by 100 for better presentation.

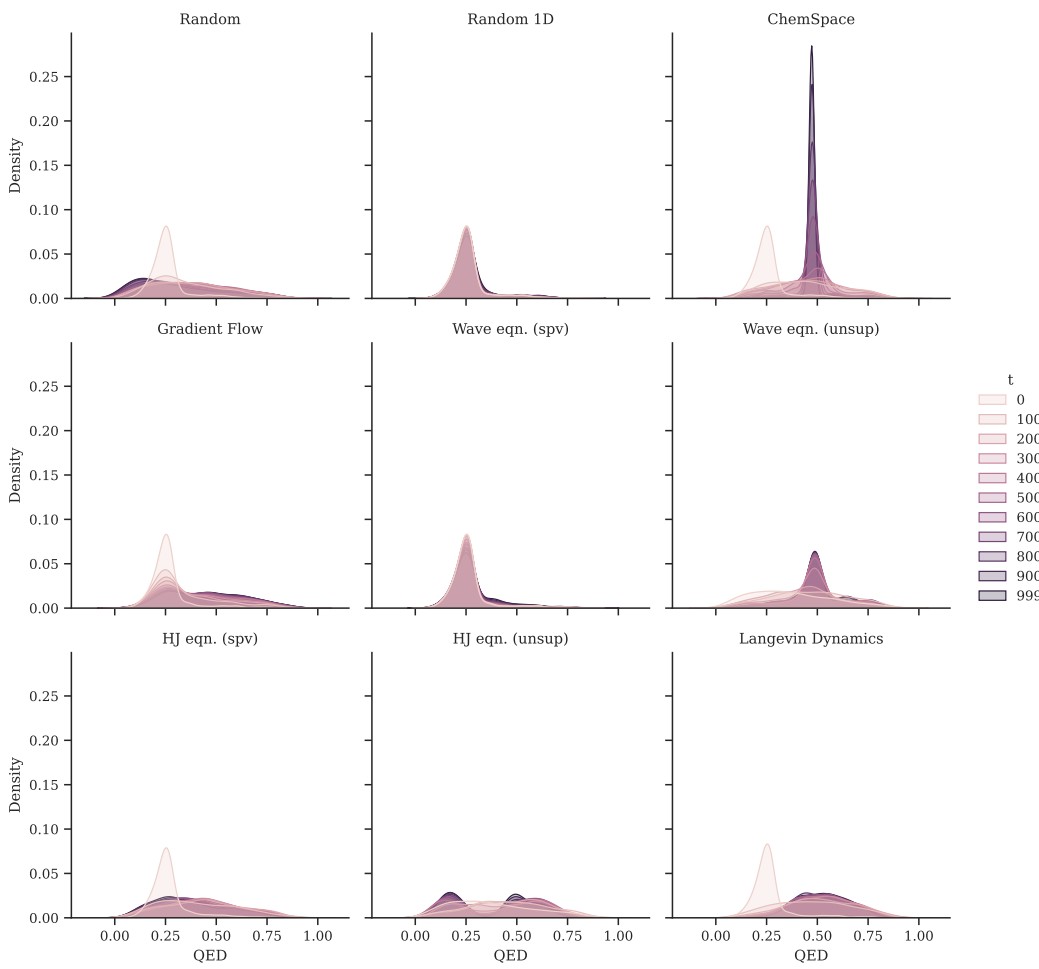

Figure 4: **Distribution shift for QED optimization** The distribution almost doesn't change during 1,000 steps of optimization, implying that the flow is close to zero almost everywhere

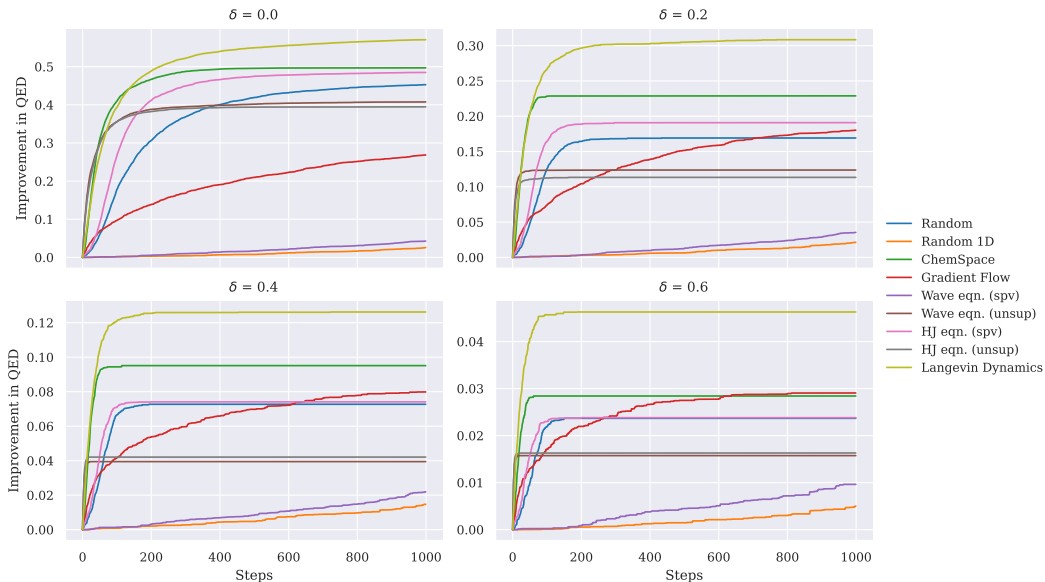

Figure 5: **Optimization Convergence** Langevin Dynamics shows faster convergence and achieves greater improvement in QED.

### F.5 MOLECULE MANIPULATION RESULTS

|  | PLOGP (↑) | QED (↑) | SA (↓) | DRD2 (↑) | JNK3 (↑) | GSK3B (↑) |
|---|---|---|---|---|---|---|
| RANDOM-1D | 0.10 / 0.80 | 0.20 / 0.20 | 0.60 / 1.10 | 0.50 / 1.50 | 10.60 / 35.10 | 0.40 / 1.10 |
| RANDOM | 5.90 / 36.30 | 7.10 / 23.10 | 7.90 / 28.20 | **10.30** / 57.60 | **14.70** / 62.50 | 9.90 / 43.10 |
| CHEMSPACE | 6.60 / 21.20 | 7.70 / 21.40 | 6.80 / 24.20 | 5.40 / 69.50 | 10.70 / **64.50** | 10.60 / 35.50 |
| WAVE (UNSUP) | 4.20 / 23.70 | 6.00 / 22.00 | 4.30 / 18.10 | 7.10 / 68.60 | 7.30 / 33.20 | 5.00 / 15.00 |
| WAVE (SPV) | 7.30 / 40.20 | 4.60 / 15.00 | 8.70 / 28.30 | 2.30 / 19.40 | 4.30 / 38.50 | 1.70 / 21.00 |
| HJ (UNSUP) | 7.30 / 38.80 | 6.00 / 21.20 | 1.90 / 7.40 | 8.00 / **71.10** | 11.10 / 55.80 | 8.90 / **45.40** |
| HJ (SPV) | 3.00 / 21.60 | 6.60 / 20.70 | 7.40 / 28.00 | 3.90 / 33.20 | 5.60 / 20.00 | 3.20 / 16.20 |
| GF (SPV) | 11.10 / 44.50 | **12.40** / **27.50** | 14.30 / **41.00** | 6.60 / 23.70 | 11.80 / 35.20 | **13.70** / 41.20 |
| LD (SPV) | **11.20** / **45.30** | 11.40 / 26.80 | **14.40** / 40.40 | 1.70 / 10.10 | 7.40 / 24.60 | 8.40 / 38.20 |

Table 5: **Success Rate of traversing latent molecule space to manipulate over a variety of molecular properties.** Numbers reported are strict success rate/relaxed success rate in %. (SPV denotes supervised scenarios, UNSUP denotes unsupervised scenarios).

### F.6 MOLECULE MANIPULATION

Molecule manipulation is a relatively new task proposed in Du et al. (2023) to study the performance latent traversal methods. Specifically, the main idea of molecule manipulation is to find smooth local changes of molecular structures that simultaneously improve molecular properties which is essential to help chemists systematically understand the chemical space. As our methods are general in both learning the traversal direction with surrogate model (supervised) or with only structure change (unsupervised), we evaluate below the performance of both use cases.

**Supervised Molecule Manipulation** Table 5 shows the success rate results of manipulating 1,000 randomly sampled molecules to optimize each desired property. Following Du et al. (2023), we traverse the latent space for 10 steps in the traversal direction of each method as reported the strict and relaxed success rate. Details of the definition of these metrics can be found in Appendix F.2. Among all the approaches, our method with gradient flow achieves the highest success rates on multiple properties such that it takes the steepest descent of the surrogate model. When the step size is small enough, it is reasonable to learn a smooth path. However, the results vary across properties. One particular example is DRD2 such that supervised approaches all fail to achieve good success rates.

**Unsupervised Molecule Manipulation** As the correspondence between specific molecular properties and learned latent potential flows is not explicitly given in the unsupervised scenario, we use

an artificial process to mimic the use case in reality. Specifically, we learned 10 different potential energy functions representing 10 disentanglement potential flows following Algorithm 2 using Wave equation and Hamilton-Jacobi equation and validated them on 1,000 unseen molecules. For each potential flow, we measure the real properties of molecules generated along the manipulation trajectory of 10 steps. We report the average of all Pearson correlations between the sequence of properties and time step sequence from 1 to 10 in Table 2 for Wave eqn. and Table 3 for HJ eqn. The best correction potential flow is selected for each property representing the learned Jacobian structural change that would most effectively optimize the corresponding property.

In Table 5, we can observe that even though it is without supervised training of traversal directions, the flow still learns meaningful directions from molecular structure to property changes. Surprisingly, the results of manipulating molecules for DRD2, JNK3, and GSK3B in unsupervised settings are better than in supervised settings. We hypothesize that this is partially because of the training and generalization errors of the surrogate model, as shown in Appendix Figure 3. On the contrary, the structure change measurement does not provide supervision but is reliable. We would like to point out that this is an open question in chemistry, often referred as to the structure-activity relationship (Dudek et al., 2006), such that it is important to know the correspondence between structure and activity. We believe this is a promising result to demonstrate that generative models "realize" molecular property by learning from structures.

Among the quantitative results, it is notable that the random direction achieves surprisingly high success rates, we argue that's because of the specific property of the learned latent space. The underneath generative model regulated the latent space to be smooth such that similar molecular structures are often mapped to close areas in the latent space. We also analyze in Appendix G such that we find some molecular properties are highly correlated with their latent vector norms in which a random direction always increases the norm and therefore successfully manipulates a portion of molecules by chance.

## F.7 ADDITIONAL MOLECULE OPTIMIZATION RESULTS

| METHOD | PLOGP | | | QED | | |
|---|---|---|---|---|---|---|
| | 1ST | 2ND | 3RD | 1ST | 2ND | 3RD |
| RANDOM | 4.13 | 3.50 | 3.37 | 0.933 | 0.931 | 0.927 |
| CHEMSPACE | 3.91 | 3.76 | 3.69 | 0.929 | 0.925 | 0.921 |
| LIMO (REPROD.) [1] | 4.13 | 4.04 | **4.00** | 0.939 | **0.936** | **0.933** |
| WAVE (SPV) | 3.06 | 2.97 | 2.48 | 0.933 | 0.933 | 0.931 |
| WAVE (UNSUP) | 3.51 | 3.24 | 3.14 | 0.932 | 0.932 | 0.931 |
| HJ (SPV) | 3.24 | 3.09 | 2.68 | 0.943 | 0.928 | 0.927 |
| HJ (UNSUP) | -0.80 | -1.39 | -1.52 | 0.932 | 0.931 | 0.931 |
| LD | **4.33** | **4.07** | 3.99 | **0.941** | 0.935 | 0.932 |

Table 6: **Unconstrained plogP and QED maximization.** All results are produced on the same VAE pre-trained on the ZINC250k dataset. (SPV denotes supervised scenarios, UNSUP denotes unsupervised scenarios).

**Baselines.** For molecule optimization, we follow the same experiment procedure as in Eckmann et al. (2022)[2]. To ensure a fair comparison, we use the same pre-trained VAE model for all the methods. The details about the baselines are reported in Appendix F.2.1.

**Unconstrained Molecule Optimization.** For unconstrained optimization problems, we randomly sample 100K molecules from the latent space and report the top 3 scores after 100 steps of manipulation of each method in Table 6. All methods use 0.1 relative step sizes for fair comparison. Among them, we observe that gradient flow and Langevin dynamics are the two best approaches that find better candidates than other methods. This is reasonable since they are taking the steepest descent of the energy function in Euclidean and probability space, respectively. However, we notice that in

---

[1]The original method was trained on a much larger training set (2M vs 250K), we report the reproduced results for fair comparison. We also use SGD instead of Adam to align with our scheme.

[2]Note that we notice there is a misalignment of normalization schemes for the plogP property in the previous literature, so we only rerun and compare with related methods that align with our normalization scheme. Details can be found in Appendix F.2.

Figure 2, Langevin dynamics pushes the entire distribution much further than gradient flow despite the optimal solutions are similar. In addition, we find that despite the random direction seeming to be effective in optimizing molecules, it does not push the distribution much.

## F.8 MULTI-OBJECTIVE OPTIMIZATION RESULTS

| δ | Random | Random-1D | ChemSpace | Gradient Flow | Wave (SPV) | Wave (UNSUP) | HJ (SPV) | HJ (UNSUP) | LD |
|---|---|---|---|---|---|---|---|---|---|
| | | | | | **QED** | | | | |
| 0 | 45.5 ± 13.3 (99.5) | 13.0 ± 12.7 (13.6) | 47.0 ± 12.9 (99.6) | 31.9 ± 17.9 (89.9) | 14.6 ± 14.4 (26.8) | 39.0 ± 18.5 (96.1) | 45.2 ± 13.7 (98.9) | 40.6 ± 19.4 (96.8) | **47.0 ± 13.1 (99.6)** |
| 0.2 | 20.7 ± 14.4 (81.8) | 11.2 ± 11.0 (12.6) | 25.9 ± 16.7 (88.2) | 23.0 ± 16.4 (80.0) | 12.3 ± 11.7 (24.2) | 18.3 ± 14.0 (69.4) | 22.7 ± 15.6 (84.9) | 15.5 ± 12.8 (71.4) | **27.6 ± 16.3 (92.1)** |
| 0.4 | 12.8 ± 10.7 (57.0) | 9.3 ± 8.7 (10.8) | **15.5 ± 13.2 (63.4)** | 14.4 ± 12.6 (59.4) | 9.8 ± 10.0 (19.8) | 10.0 ± 9.6 (43.2) | 13.9 ± 12.0 (57.5) | 9.2 ± 9.4 (46.2) | 15.4 ± 12.5 (71.4) |
| 0.6 | 8.0 ± 8.3 (29.5) | 6.9 ± 6.7 (5.8) | **9.7 ± 10.1 (31.6)** | 9.4 ± 9.1 (32.2) | 6.8 ± 6.5 (12.4) | 6.4 ± 7.0 (24.9) | 9.3 ± 9.7 (30.2) | 6.4 ± 7.3 (26.9) | 9.6 ± 9.4 (40.1) |
| | | | | | **SA** | | | | |
| 0 | 8.34 ± 8.06 (37.2) | 6.49 ± 7.06 (10.9) | 7.92 ± 7.71 (42.8) | 9.56 ± 8.49 (44.0) | 6.37 ± 6.30 (12.9) | 15.21 ± 10.17 (89.2) | 8.93 ± 8.39 (52.1) | 16.03 ± 10.31 (91.0) | **11.51 ± 10.44 (69.2)** |
| 0.2 | 6.70 ± 7.11 (27.0) | 6.47 ± 7.10 (10.6) | 6.63 ± 6.71 (36.1) | 7.19 ± 6.66 (35.8) | 5.77 ± 5.81 (12.4) | 7.69 ± 6.51 (70.8) | 6.69 ± 6.61 (38.8) | 7.35 ± 6.34 (68.5) | **7.51 ± 7.41 (45.4)** |
| 0.4 | 4.80 ± 5.73 (19.1) | 5.06 ± 4.95 (9.4) | **4.75 ± 5.45 (25.2)** | 5.27 ± 5.30 (26.6) | 4.54 ± 4.51 (10.5) | 3.92 ± 3.86 (45.9) | 5.21 ± 5.72 (29.2) | 4.22 ± 4.09 (46.8) | 4.50 ± 4.95 (33.6) |
| 0.6 | 2.61 ± 3.21 (10.9) | 3.83 ± 3.73 (6.0) | **2.89 ± 3.22 (15.4)** | 3.04 ± 3.62 (17.1) | 3.44 ± 3.59 (7.1) | 2.22 ± 1.97 (25.9) | 3.23 ± 3.40 (18.6) | 2.73 ± 2.85 (27.8) | 2.75 ± 3.09 (20.1) |

Table 7: **Similarity-constrained Multi-objective (QED+SA) maximization.** The value of QED and SA is scaled to both have a range from 0 to 100 for an equal-weighted sum. The method with the highest equal-weighted sum score of QED+SA of each structure similarity level is bolded.

**Multi-objective Molecule Optimization.** As we are learning distinct vector fields and potential energy functions for each property, they can be readily added together for multi-objective optimization (Eckmann et al., 2022; Du et al., 2023). To generate molecules that are optimized on multiple properties, we use a similar setting as similarity-constrained molecule optimization to select 800 molecules from the ZINC250k dataset with the lowest QED and aim to generate molecules with high QED as well as SA simultaneously. At each time step, the latent vector is optimized following the averaged direction of two supposed potential flow directions. This scheme could be seamlessly generalized to $m$-objectives optimization and is commonly used in related works. Table 7 shows that Langevin dynamics and ChemSpace achieve the best or competitive performance at all similarity cutoff levels.

## G LATENT SPACE VISUALIZATION AND ANALYSIS

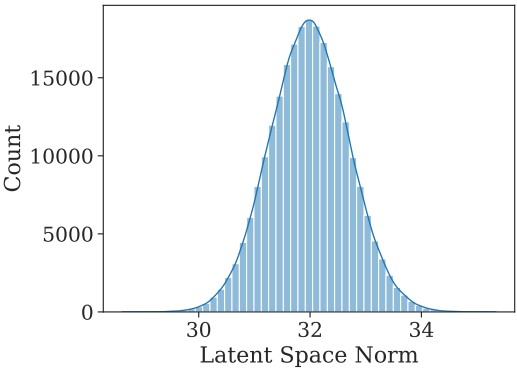

Figure 6: **Latent Vector Norm.** Distribution of the norm of the latent vectors projected from training dataset onto the learned latent space.

As observed in experiments such that random directions perform surprisingly well on molecule manipulation and optimization tasks, we look into the learned latent space to understand its structure. As the prior of a VAE is an isotropic Gaussian distribution, we first verify if the learned variational poster also follows a Gaussian distribution and we find that it does learn so from the evidence shown in Figure 6, where the norm of the molecule projected to the latent space concentrate around 32 which is around $\sqrt{d}$ such that the latent dimension $d$ is 1024. We also visualize in Figure 7 that how the properties of the molecules in the training dataset are related to their latent vector norms. Surprisingly, we find a strong correlation between almost all molecular properties and their latent norms. Combining this two evidences, it is not surprising that a random latent vector taking a random

direction will change the molecular property smoothly and monotonically. In addition, we further plot when we traverse along a random direction in the latent space, how the change of the norm may correspond to the change of a certain property. Among them, we find that SA is particularly in strong positive correlation with the traversal in Figure 8. Though the emergence of the structure in the latent space is interesting and suggests that better algorithms can be developed to exploit the structure, we leave this to future work.

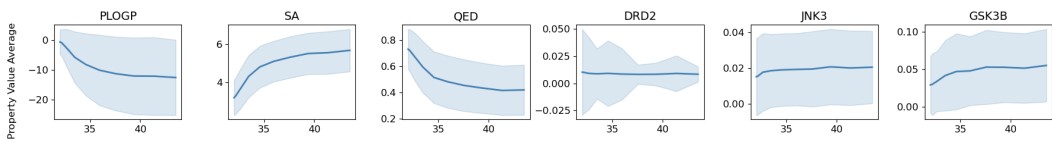

Figure 7: **Embedding Norm against Property Value of each path.** Norm and property value of molecules along the direction of latent traversal with a random direction. The middle curve shows the mean property value and latent embedding norm for all paths. The shaded area is the standard deviation of property value.

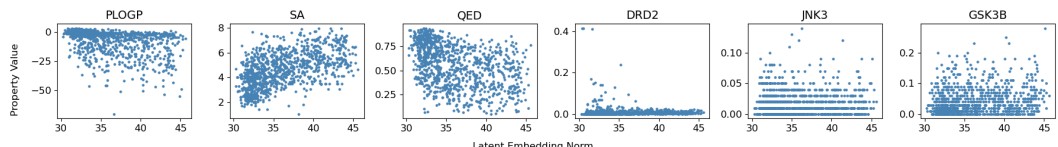

Figure 8: **Embedding Norm against Property Value of each Molecule.** Scatter plot of norm and property value of individual molecules in the training set encoded in the latent space.

## H  QUALITATIVE EVALUATIONS

In addition to quantitative evaluations, we demonstrate some qualitative evaluations in this section. We showcase one selected manipulation path Figure 9 in by gradient flow (GF) on plogP. We can see the learned path follows the trend to replace the O-containing non-stable macrocycle step by step with six- or three-member rings while optimizing the desired property (i.e, plogP). For molecule optimization, we also showcase one selected path Figure 13 Fokker Planck flow on plogP. We also observe a similar trend such that it replaces the N-containing fused ring step by step with three-member rings while optimizing the desired property (i.e., plogP). Additional molecule manipulation paths are shown in Figure 10, Figure 11, Figure 12, and optimization paths are shown in Figure 14, Figure 15, Figure 16.

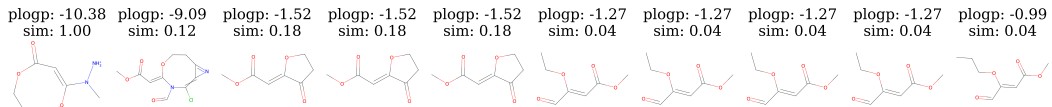

Figure 9:  **Molecule Manipulation Trajectory**. The figure shows a full 10 step manipulation by gradient flow on plogP. Each molecule in the figure represents a step in the path.

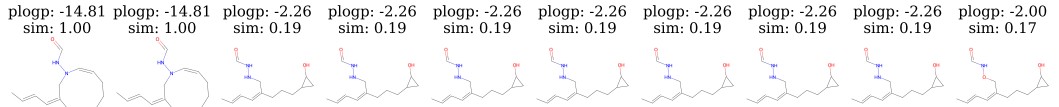

Figure 10: **Molecule Manipulation Trajectory** Molecule manipulation by random direction on plogP.

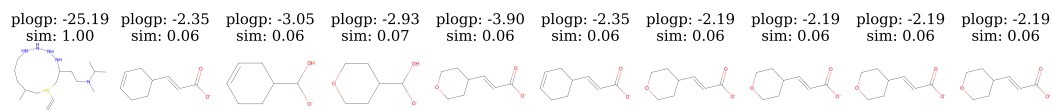

Figure 11: **Molecule Manipulation Trajectory** Molecule manipulation by Fokker Planck flow on plogP.

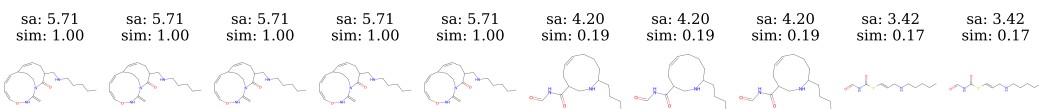

Figure 12: **Molecule Manipulation Trajectory** Molecule manipulation by unsupervised wave flow on SA.

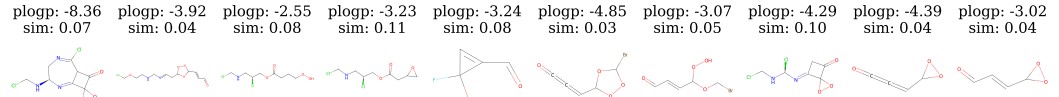

Figure 13: **Molecule Optimization Trajectory**. From left to right, each molecule is a step selected from a full 1000 step optimization trajectory by Fokker Planck flow on plogP. Only 10 intermediate steps, during which the molecules underwent changes, are shown in this gifure.

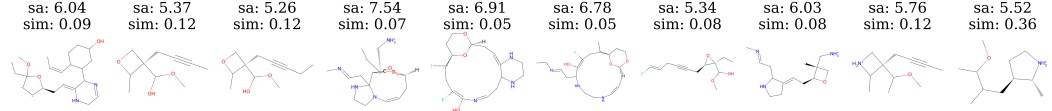

Figure 14: **Molecule Optimization Trajectory** Molecule optimization by random direction on SA.

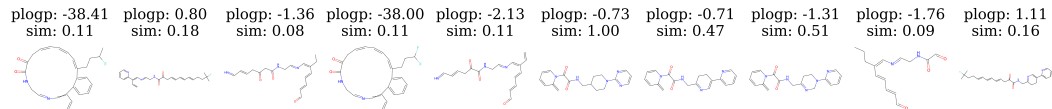

Figure 15: **Molecule Optimization Trajectory** Molecule optimization by unsupervised gradient flow on plogP.

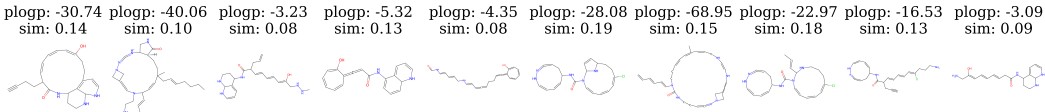

Figure 16: **Molecule Optimization Trajectory** Molecule optimization by unsupervised wave flow on plogP.

