# OpenReview forum: "Traversing Chemical Space with Latent Potential Flows"
_ICLR.cc/2024/Workshop/AI4DiffEqtnsInSci — AI4DiffEqtnsInSci @ ICLR 2024 Poster_

### Official Review · Reviewer_VD6q · 2024-02-19
**The authors consider the problem of studying and exploring the dynamics of the (chemical) latent space from a generative model.**

**Rating:** 6
**Confidence:** 2

**Review:**

The authors consider the problem of studying and exploring the (chemical) latent space of a generative model. They aim to make connections of how one can explore the latent space of a (generative model) and dynamical systems. They use different approaches e.g. computing the gradient (gradient ascent or descent), learn a wave type parametrization of the potential (with the help of PINNS) a HJE and also use Langevin dynamics.

In my comments below I would mostly focus on the dynamic’s aspect of the work and less on the chemical/molecular aspect that is not my field.

I believe this is a sound paper and that the authors have done a lot of work.

The four-page limitation makes the paper difficult to follow therefore going back and forth between the main text and the appendix becomes necessary to get understand the overall frameworks. There are a few parts of the paper that I believe need improvement, I provide more details below.

Major Comments:

(I)	In many expressions in the main text there is a superscript k e.g. (equations 3,5,6) I do not see where this is defined. Can you clarify/change this?

(II)	Can the authors mention why learning a vector field is better and not evaluating a network in a dense grid and choosing the minimum or maximum on the grid? What cases one can expect the one to be better than the other?

(III)	What happens if the maximum, minimum is close to the boundary? Are the trajectories going to “leave” the latent space and explode? Are there ways they can encode this information to their model?

(IV)	Why the Langevin dynamics might be better? Is it better regardless of the size of the noise?

(V)	Even though I understand the limitations because of the four-page limit I think is important to improve the section “conclusions, limitations”. Now is more a reiteration of the overall paper and it does not mention what the conclusions and the limitations are.


Minor:

Abstract: (a) Check a little bit the English (e.g. that unified -> unifies?)

Introduction: (a) (Figure 1) The NN in the schematic of the decoder has as input two neurons and one (yellow) output, I think this should have been reversed right? (from less inputs to more outputs in the decoder).

(b) Perhaps also a change is needed for the green box in the schematic for the predictor (go from the yellow to a function)?

(c ) The yellow color is not so visible.

Background: (Traversing Latent Space of Molecules) (a) The gradient of the potential energy function h is estimated in equation 1. Can you say how this is computed (automatic differentiation, symbolic differentiation, finite differences)? The same holds also for the case of the potential φ later (e.g. equation 3).

(b) The first time PINNs are mentioned is in page 2 perhaps move the citation for the paper there?

(c ) “Compared with traditional PDE solvers PINNs can be orders of magnitude faster”; can you clarify if this includes both the training time or only the inference time?

(d) Please clarify for each case you used PINNs if you solve the forward or inverse problem (fit the constant c)?

(e ) Do you assume in this case you have some data for the potential or not? If not, how can you guarantee the uniqueness of the solution for the constant?

(f) Equation (6) does not mention a loss function for the Boundary Conditions, is this necessary or not in your case if not can you please explain why?


Supervised semantic potential guide (a) The input for the surrogate model $h_η$ lives in the high dimensional space is this correct? I thought we are working with the latent space of the generative model.


(b) It is not clear to me, how equation 7 gives you the vector field z_t, can you add a few words in the SI and explain in more detail if you think is important for your overall scheme? The output seems to be just a scalar to me.

(c ) In high-dimensional space, a trivial solution exists such that all the flows learn the exact direction -> Can you justify that why this is the case, intuitively I would expect the opposite?

(d) Table (1) What is delta? Please explain>


Appendix :
(a) Disentanglement Regularization: Since here there is no limit in the SI can you make things more clear? Why having the maximum Jacobian change is a problem? Why an auxiliary classifier is helpful? Perhaps cite or say a few words here of what is auxiliary classifiers.

(b) Check the text after equation 28, I think it might be wrong.

(c ) Algorithms: For the Langevin dynamics how β was selected. How sensitive the method might be to that value?  Can you mention include the selected parameters in your test cases?

(d) For algorithms 2, I don’t know if I missed it in the text or in the Appendix what sampling the timestamps and the potential functions means and why there are two random variables for that?

(e ) Check Caption for Figure 4 in the SI.

(f) Figure 9, those are representative molecules, right since you have a generative model. Perhaps check the captions.

---

### Meta-Review · Area_Chair_hVFJ · 2024-03-01

**Recommendation:** Accept (Poster)

**Metareview:**

Thanks to the reviewer for the detailed review and feedback.
The paper proposes a generative modeling framework (VAE based) , Chemflow, for chemical molecule structure analysis and manipulation, Authors are expected to go through all comments and address them clearly in their final revision. Under this condition, the work can be accepted as poster.

---

### Decision · Program_Chairs · 2024-03-02

Accept (Poster)